# Identification of Key Areas for Ecosystem Restoration Based on Ecological Security Pattern

Jiaquan Duan , Xuening Fang, Cheng Long , Yinyin Liang, Yue 'e Cao *, Yijing Liu and Chentao Zhou

College of Environmental and Geographic Sciences, Shanghai Normal University, Shanghai 200234, China
* Correspondence: caoyuee@shnu.edu.cn

**Abstract:** Ecosystem degradation and conversion are leading to a widespread reduction in the provision of ecosystem services. It is crucial for the governance of regional land spaces to rapidly identify key areas for ecosystem restoration. Herein, we combined the InVEST Habitat Quality Model with the granularity inverse method to identify ecological sources in Jiashi county, China, based on the "source-corridor" ecological security pattern paradigm. The minimum cumulative resistance model and circuit theory were adopted to diagnose the ecological "pinch points", barrier points, break points, and key restoration areas for land space. Our results show that: (1) the area of the ecological source and the total length of the ecological corridor were identified as 1331.13 km$^2$ and 316.30 km, respectively; (2) there were 164 key ecological "pinch points" and 69 key ecological barrier points in Jiashi county, with areas of 15.13 km$^2$ and 14.57 km$^2$, respectively. Based on the above ecological security pattern, recovery strategies are put forward to improve regional ecosystem health. This study describes the best practices which can be used to guide the planning and implementation of ecosystem restoration at the local landscape scale.

**Keywords:** ecological security pattern; ecological restoration of land space; InVEST model; granularity inverse method; circuit theory

## 1. Introduction

Ecological restoration is the process of helping destroyed or degraded ecosystems restore their original structure and function [1,2]. It can reverse land degradation, enhance biodiversity, and facilitate the delivery of crucial ecosystem functions [3]. However, not all ecosystem management can support ecosystem restoration, and some research indicates that widespread tree planting in northwest China may have made the water shortage even worse [4]. The primary goal of ecological restoration research is the systematic identification of critical regions for ecological restoration. Consequently, ecological restoration necessitates a methodical and comprehensive evaluation of the effects of various causes, which is also the subject of emphasis in Chinese ecological restoration practice. The Chinese government has proposed the implementation of an ecological territory restoration plan as part of the new round of governmental institutional reforms. As a result, China has moved from managing a single ecological restoration project to a systematic and all-encompassing management of ecological territory restoration. The scientific and governmental sectors are currently very concerned about how to address the systemic and comprehensive nature of ecological restoration, and how to systematically identify the essential areas for ecological restoration of territory land space.

Various Chinese scholars have adopted the ecological security pattern as the core research framework to identify key regions for ecological restoration, using circuit theory and other techniques to systematically identify important areas for ecological restoration in territory land spaces [5–7], while the majority of international researchers use ecological networks as their study framework, placing a higher priority on protecting the environment and organisms [8,9]. For instance, Hofman et al. suggested a technique for building ecological

networks that can serve as a useful guide for the protection of biodiversity [10]. Use of the ecological security pattern combined with circuit theory is considered one possible approach that can reflect the methodical and all-encompassing nature of ecological restoration in land space. Yu's team initially put forth the ecological security pattern paradigm [11], which was steadily developed into the "source-corridor" paradigm [12]. The "source-corridor" paradigm identifies ecological sources, constructs ecological resistance surfaces, and, finally, constructs ecological corridors based on ecological sources and ecological resistance surfaces. Ecological sources can be identified by extracting ecological land from land-use/cover data [13,14] and assessing the functional importance and sensitivity of regional ecosystem services [15,16], landscape connectivity [9,17,18], the trade-offs of ecosystem services [19,20], and supply and demand for ecosystem services [21–24]. A minimum resistance model can be used to build ecological corridors, which requires the creation of ecological resistance surfaces. While most studies directly assign values based on the type of land use/cover, some studies also used nighttime lighting data, the slope topography factor [14], hazard sensitivity [15], and other data to make corrections on this basis.

However, the following issues can arise within the "source corridor"-based ecological security pattern research paradigm: (1) Most studies evaluate additional ecological source extraction based on land use/cover data in terms of biodiversity and connectivity, ignoring the ecological source challenges observed in actual conservation. The future conservation of ecological sources can benefit from the identification of ecological sources based on the most recent land use/cover data from the third national land survey in China. (2) The use of a single analysis of landscape connectivity for the extraction of ecological sources cannot capture their full functionality, and it poses great challenges for the identification of the crucial role of ecological sources in regional ecosystems. Other characteristics of the landscape pattern, such as patch density and patch cohesion, can influence the landscape function. The granularity inverse method incorporates the effects of different landscape indices, which can indicate the overall connectivity of the sources. (3) Directly using land type assignments to determine ecological corridors creates a resistance surface that cannot adequately capture the influence of other elements, such as human activities and environmental conditions. Furthermore, only a few studies have examined the variation in the combined influence of both human activities and natural elements on ecological resistance surfaces for homogeneous land use/cover types, despite the fact that various scholars have identified modifications of resistance surfaces by topography or human activities.

Jiashi county, in the Kashi region (in which the Xinjiang Production and Construction Group is based), is a significant border city in western China that is well-known for its fruit and melons. Since the Western Development Strategy was put into place, Jiashi county's economy has been growing quickly. However, the need for environmental protection is also important. How economic growth and environmental protection can be balanced in Jiashi county has emerged as a significant problem that needs to be resolved during the 14th Five-Year Plan period. This study aimed to (i) combine the InVEST Habitat Quality Model with the granularity inverse method to identify ecological sources in Jiashi county, China, based on the "source-corridor" ecological security pattern paradigm; (ii) adopt the minimum cumulative resistance model and circuit theory to diagnose the ecological "pinch points", barrier points, break points, and key restoration areas for land space; and, finally, (iii) propose strategies for ecosystem restoration.

## 2. Material and Methods

### 2.1. Study Area

Jiashi county is located in the northern part of Kashi region, Xinjiang Uyghur Autonomous Region, with geographical coordinates $39°16'\sim40°00'$ N and $76°20'\sim78°00'$ E (Figure 1). The total area of the county is 6528 km$^2$. It has a temperate continental arid climate, with hot summers, cold winters, and scarce precipitation. Jiashi county's sandy region has gradually grown due to human economic activity, and the majority of the area has been affected by sand wind. The accumulation of these issues has endangered the

ecological security of Jiash county, and has become a significant barrier to the county's sustainable development.

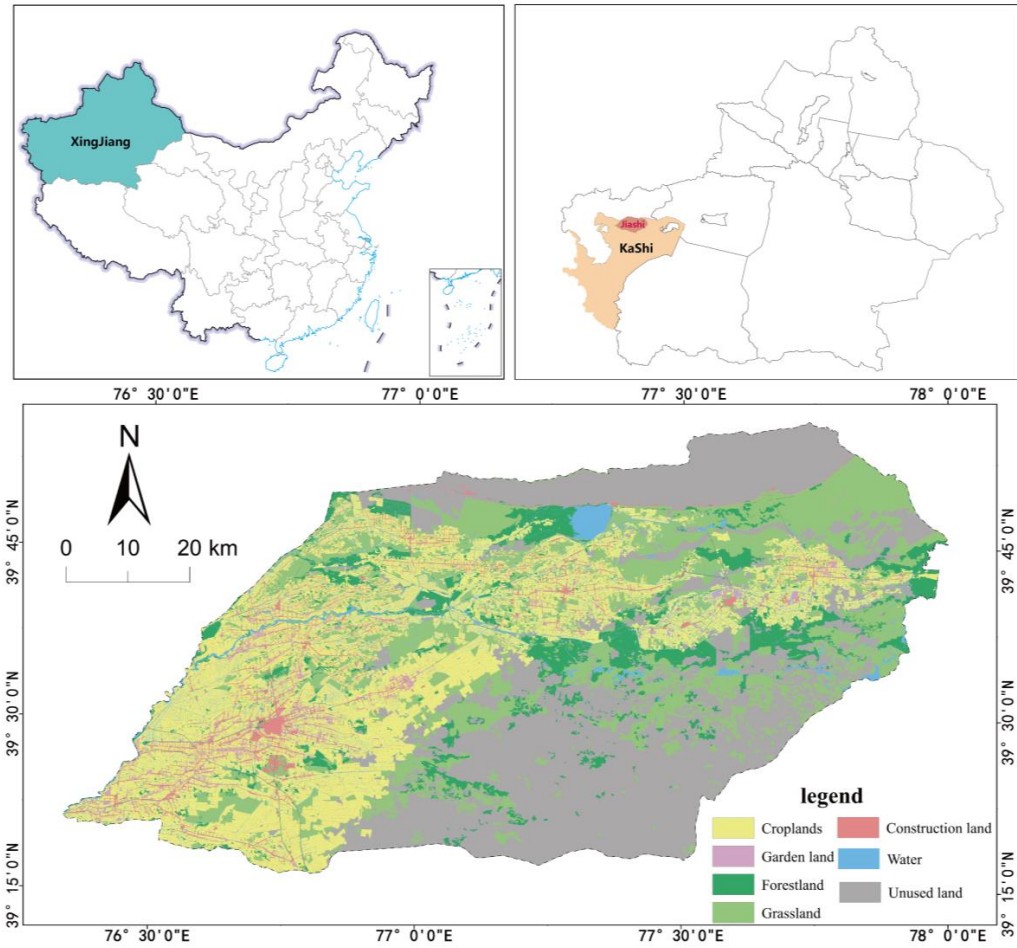

**Figure 1.** Location Map and Land Use Map of Jiashi County.

### 2.2. Datasets

The current land use/cover data of the Third National Land Survey were from the Natural Resources Bureau of Jiashi county. The Third National Land Survey is the third national land survey in China. It is a significant assessment of the state of the territory, and serves as the country's foundation for creating essential strategic plans and policy directives for economic and social advancement. The digital elevation model (DEM) was obtained from The Geospatial Data Cloud Platform, Available online: http://www.gscloud.cn (accessed on 25 June 2021) with a spatial resolution of 30 m; the nighttime lighting data were obtained from the National Geophysical Data Center NGDC under NOAA of the National Oceanic and Atmospheric Administration (National Geophysical Data Center, Available online: http://www.ngdc.noaa.gov/dmsp/download.html (accessed on 26 March 2022).

### 2.3. Study Flow Chart

The research method includes three parts, i.e., the identification of the ecological source, the construction of resistance surfaces, and the identification of ecological restoration patterns in land space. The identification of the ecological source is conducted by first extracting ecological land from land use/cover data, then calculating landscape connectivity and habitat quality based on ecological land using the granularity inverse method and InVEST model, and, finally, extracting ecological sources from ecological land by integrating connectivity and habitat quality. A uniform land use type resistance surface is first generated using a land cover map; this surface is then modified utilizing information

on nighttime illumination and topography to create the final, fully developed resistance surface. The minimum resistance model and circuit theory are then used to extract significant ecological corridors and key areas of ecological restoration from the modified resistance surface and ecological sources, and, hence, to build a spatial ecosystem restoration pattern for the composition map (Figure 2).

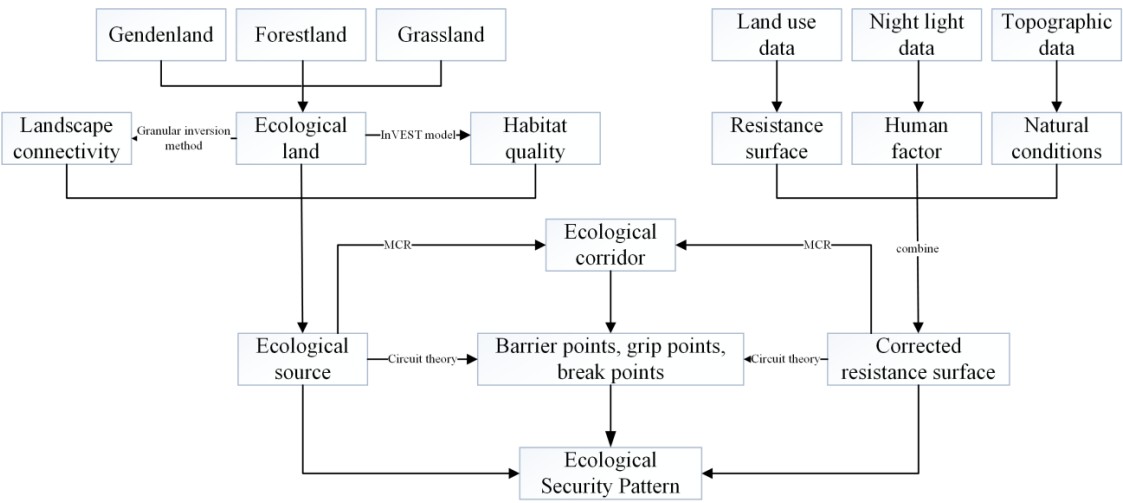

**Figure 2.** Study flow chart of creating an ecological security pattern.

*2.4. Identification of Ecological Sources*

2.4.1. Extraction of Ecological Land

Ecological sources are habitat patches that are crucial for regional ecological security, because they play a significant role in regional ecological processes and functions, as well as providing significant radiative functions [12]. Finding ecological sources is crucial for developing ecological security patterns and becoming geared for ecological restoration [16]. Rivers, wetlands, and marshes are not considered ecological sources for extraction based on the studies of Jijun [14] et al., as there are not many rivers and marshes in Xinjiang, particularly in the southern portion of the study area, and since rivers often have seasonal water breaks. Additionally, the garden land in the study area is typically large and concentrated, offering important ecosystem service activities. These functions play a significant positive role in the conservation of biodiversity, and meet the criteria for the selection of ecological sources, which are vital to regional ecological processes and functions. Thus, forest land, garden land, and grassland were extracted from the land use/cover data to screen the preliminary ecological land as the patch for the next step of ecological source extraction.

2.4.2. Ecological Land Connectivity and Holistic Screening

By measuring the landscape pattern index at various granularities, the granularity inverse method, which is based on the principle of proof by contradiction, determines the ideal landscape component. Based on this landscape component characteristic, the method chose the ecological sources with higher connectivity [25]. According to the previous research [1,17], the nine indices of Number of Patch Components (NC), Patch Density (PD), Maximum Number of Component Plagues (Max NC), Proximity Mean Distance (PROX_MN), Proportion of Like Adjacencies (PLADJ), Connectivity Index (CONNECT), Patch Cohesion Index (COHESION), Aggregation Index (AI), and Landscape Division Index (DIVISON) were selected to characterize the landscape component structure (Table 1). After conducting numerous simulations and taking into account lessons learned from earlier research, the study discovered that the optimal granularity is typically less than 2000 m [17,18]. The calculated granularity will not display after a specific granularity, at which point the corresponding crystallinity index and connectivity index are both 0, rendering them incomparable. This can be achieved by setting the proximity index and

connection index thresholds to 2000 m. The granularity was set at 50 m due to the study area's small size. The values of both the proximity mean distance and connectivity indices are zero after 2000 m granularity, and there is also no reference value after this point. Meanwhile, referring to the study conducted by Yu et al. [17,18], approximately 21 scales can better address the trend of abrupt size change; thus, this study set 50 m, 100 m, 200 m, 300 m, 400 m, 500 m, 600 m, 700 m, 800 m, 900 m, 1000 m, 1100 m, 1200 m, 1300 m, 1400 m, 1500 m, 1600 m, 1700 m, 1800 m, 1900 m, and 2000 m at all 21 scales. The landscape pattern index shifts along with the granularity, and the connection and integrity of the matching granularity are at their highest when the overall landscape pattern index changes drastically in trend. Thus, the aforementioned nine indices are calculated using principal component analysis. The principal component function identifies the ideal structure for the composition of the landscape and characterizes the ecological source's overall structural qualities.

**Table 1.** Landscape pattern index and its significance.

| Indicator Name | Formula | Description | Ecological Significance |
|---|---|---|---|
| Number of Patch Components (NC) | N/A | The spatial connectivity of different patches of a specific landscape type is expressed in two interrelationships, connected and unconnected, and the connected patches form a structurally and functionally interconnected whole, i.e., a landscape component. | The interconnected patches in the region are one component. |
| Patch density (PD) | $PD = N_i / A$ | $N_i$ is the total area of the $i$th landscape type; A is the total area of the landscape. | The patch density characterizes the number of the patch within a specific area, reflecting the specific degree of the patch. |
| Proximity mean distance (PROX_MN) | $PROX\_MN = \sum_{i=1}^{n} \frac{a_{ij}}{h_{ij}^2} / n$ | $a_{ij}$ is the adjacent area of patch $i$ and central patch $j$; $h_{ij}$ is the shortest distance between patch i and central patch $j$. | The average proximity distance reflects the distance of the patch from the center. |
| Proportion of like adjacencies (PLADJ) | $PLADJ = \left( \frac{g_{ij}}{\sum_{k=1}^{m} g_{ik}} \right)$ | $g_{ij}$ is the number of focal inclusions between patch i and neighboring patch $j$; $g_{ik}$ is the number of focal inclusions between patch $i$ and all neighboring $k$ patches. | The adjacency ratio is a metric that analyzes the degree of aggregation between cells from a holistic perspective, by viewing the components spatially as scattered cells. |
| Connectivity index (CONNECT) | $CONNECT = \frac{\sum_{k=1}^{n} c_{ijk}}{\frac{n_i(n_i-1)}{2}} \times 100$ | $c_{ijk}$ is the connectivity status of patches $j$ and $k$ associated with patch type $i$ within a critical distance; $n_i$ is the number of patches of patch type $i$ in the landscape. | Connectivity reflects the functional connectivity between landscape components. |
| Patch cohesion index (COHESION) | $COHESION = 100 \left[ 1 - \frac{\sum_{j=1}^{n} p_{ij}}{\sum_{j=1}^{n} p_{ij} \sqrt{a_{ij}}} \right]^{-1}$ | $p_{ij}$ is the perimeter of patch $j$ of landscape type $i$; $a_{ij}$ is its area; A is the total number of grids in the landscape. | The cohesion index reflects the natural connectivity of the patch types. |
| Landscape division index (DIVISION) | $DIVISION = \left[ 1 - \sum_{j=1}^{n} \left( \frac{a_{ij}}{A} \right) \right]$ | $a_{ij}$ is the area of patch $j$ of landscape type $i$; A is the total area of the landscape. | The sub-dimension reflects the proportion of patch area to the total landscape area, reflecting the extent to which the landscape is divided in space. |
| Aggregation index (AI) | $AI = \frac{g_{ij}}{max g_{ij}}$ | $g_{ij}$ is the common edge length of patch $j$ of landscape type $i$. | Aggregation reflects the degree of spatial aggregation of landscape-type patches. |
| Maximum number of component plaques (Max NC) | N/A | The number of patches in the maximum fraction. | The number of patches of the largest component reflects the size and internal structure of the largest ecological source. |

2.4.3. Biodiversity Evaluation of Ecological Land

The Habitat Quality module of the InVEST model (Integrated Valuation of Ecosystem Services and Tradeoffs), jointly developed by Stanford University, WWF, and The Nature Conservancy, can assess habitat quality and degradation in the subsurface of different ecological sources. It accomplishes this by calculating habitat quality based on the quality of different habitats and the degree of impact of different threat sources on different habitats [20]. Habitat quality is also a measure of the biodiversity status of an ecosystem, indicating the ability of the ecosystem to provide living conditions for organisms [21]. The habitat quality module generates results in the range of 0–1 [2], and the results are continuous variables; the higher the value, the higher the habitat quality. The calculation of habitat quality in the InVEST model is mainly expressed by analyzing the extent and degradation of the relevant land use/cover type and its certain vegetation type or habitat type, i.e., by threat factor data, threat source data, and land use data. The calculation formula is as follows:

$$D_{xj} = \sum_{r=1}^{R} \sum_{y=1}^{Y_r} \left( \frac{\omega_r}{\sum_{r=1}^{R} \omega_r} \right) \times r_y \times i_{rxy} \times \beta_x \times S_{jr} \qquad (1)$$

$$Q_{xj} = H_{ij} \times \left[ 1 - \left( \frac{D_{xy}^2}{D_{xy}^2 + k^2} \right) \right] \qquad (2)$$

where $D_{xj}$ is the degree of habitat degradation of raster $x$ in habitat type $j$; $r$ is the threat source; $y$ is the number of the raster of threat source $r$, and $\omega_r$ is the weight of the threat factor; $r_y$ is the stress value of raster $y$; $i_{rxy}$ is the stress level of threat source $r$ to $x$ in raster $y$, divided into exponential and linear effects; $\beta_x$ is the accessibility level of the threat source to raster $x$; $S_{jr}$ is the sensitivity of habitat type $j$ to threat source $r$; $Q_{xj}$ is the habitat quality; $H_{ij}$ is the habitat suitability; $k$ is the half-saturation parameter, usually taking a value of 2.5.

With reference to the InVEST model guidebook [26] and related studies [27,28], the three habitat types of forest land, garden land, and grassland; their habitat suitability; the maximum influence distance and weight of the threat source factors; and the sensitivity of each habitat to the threat factors were determined based on consideration of the actual situation in the study area. Arable land, mining land, commercial land, railroad land, road land, and residential land were defined as threat source factors of the habitat, and hence formed the Habitat Quality module parameter table (Tables 2 and 3). The maximum stress distance represents the maximum distance at which each threat source can affect the study area; the weight represents the weight of the impact on habitat integrity, which is relative to other threat sources; and the type of spatial recession denotes the type of degradation caused by the threat source, which is determined by predicating whether its impact increases linearly or exponentially with increasing distance.

**Table 2.** Threat source data.

| Threat Sources | Maximum Duress Distance/km | Weights | Type of Spatial Recession |
|---|---|---|---|
| Mining land | 4 | 0.5 | exponential |
| Arable land | 1 | 0.15 | linear |
| Industrial and mining land | 3 | 0.6 | linear |
| Commercial land | 5 | 1 | exponential |
| Railroad | 2 | 0.4 | linear |
| Residential land | 3 | 1 | exponential |

**Table 3.** Relative sensitivity of different ecological land types to threat sources.

| Land Use/Cover Type | Habitat Suitability | Mining Land | Arable Land | Industrial and Mining Land | Commercial Land | Railroad | Residential Land |
|---|---|---|---|---|---|---|---|
| Forest land | 0.7 | 0.8 | 0.3 | 0.5 | 0.8 | 0.6 | 0.7 |
| Garden land | 0.4 | 0.5 | 0.3 | 0.5 | 0.7 | 0.5 | 0.7 |
| Grassland | 0.55 | 0.6 | 0.3 | 0.4 | 0.7 | 0.4 | 0.7 |

In this paper, based on the extraction of ecological land use/cover data, the optimal granularity of ecological source was initially selected and combined with principal component analysis. The habitat quality calculated by the InVEST habitat quality model was then used in ArcGIS 10.2 to calculate the mean habitat quality for each granularity component, using partition statistics. The granularity was determined by the granularity inverse method. The component with relatively higher total and mean habitat quality values was chosen as the final component for extracting the ecological source using the natural break approach. In order to ascertain the ecological source, the landscape composition structure of the final component was finally established as a point of reference.

2.4.4. Construction of Resistance Surface

Species transfer horizontally, and ecological fluxes flow between patches depending greatly on the types of land use and human activities [29]. Land use/cover type, topographic slope, and human activities are the top three factors influencing ecological resistance values (resistance to outward growth of ecological sources).

The creation of the fundamental ecological resistance surface is a prerequisite for calculating the ecological resistance coefficient $RA_i$. The influence of human activities and environmental conditions on biological migration and mobility in various land use/cover types cannot be accurately reflected by the standard method of simulating ecological resistance surface by land use/cover type. The resistance values for each land use type in this study are determined as follows: forest land, 1; paddy land, 20; dry land, 30; water, 50; rural settlement, 400; town and another building land, 500, with references to pertinent studies [14,17]. The fundamental resistance surface is adjusted using topographic and nighttime lighting data, and the calculation is as follows:

$$R_i = \frac{NL_i}{NL_a} \times \frac{Slope_i + RA_i}{Slope_a + RA_a} \times R \tag{3}$$

where $NL_i$ is the light index of grid $i$; $NL_a$ is the average light index of land use type $a$ corresponding to grid $i$; $Slope_i$ is the slope index of *grid i*; $Slope_a$ is the average slope index of land use type $a$ corresponding to grid $i$; $RA_i$ is the undulation index of grid $i$; $RA_a$ is the average undulation index of land use $a$ corresponding to grid $i$; $R$ is the basic resistance coefficient of landscape type of raster $i$ based on land use type.

*2.5. Identification of Ecological Restoration Patterns in Territory Land Space*

2.5.1. Construction of Ecological Corridors

Ecological corridors are channels that connect ecological sources [30], and they have been extended to the field of ecological security protection structures. Building ecological corridors can improve ecological source connectivity, maintain ecosystem services, and reduce ecological source fragmentation. Circuit theory is a good reaction to the migration traits of plants and animals to prevent injury, as it states that the current always travels to the place with the lowest resistance first [31], creating an ecological resistance surface and calculating ecological corridors using the properties of electric charges' haphazard wandering. The calculation formula is as follows:

$$MCR = f_{min} \sum_{j=n}^{i=m} D_{ij} \times R_i \tag{4}$$

where *MCR* is the minimum cumulative resistance value of ecological source patch *j* spreading to a point; $D_{ij}$ is the spatial distance of base plane *i* traversed by the species from ecological source *j* to a point in space; $R_i$ is the basic resistance of patch *i* to ecological processes or species movement.

2.5.2. Identification of Key Areas for Ecological Restoration of Land Space

Circuit theory can be well used to identify ecological corridors and key ecological restoration areas in ecological security patterns. Areas that are crucial for preserving the environment are known as ecological "pinch points" [32]. Applying the circuit theory to ecological "pinch points", one node (ecological source) is grounded, and the other nodes (ecological source) are connected with the same current by using the circuit theory to identify ecological "pinch points". Iterative procedures are used to determine the cumulative current value of each image element. The area with the highest current value is referred to as the ecological "pinch point" [30]. The ecological "pinch point" should be protected as a matter of priority, since it has a high current density and is irreplaceable, and its destruction or loss is likely to cut off the connection between ecological sources [33]. The "pinch points" in the ecological corridor are the areas with the highest current density, and even a slight loss in these areas could have an impact on how well the corridor connects. The "pinch points" were located, and the "all to one" computation mode was chosen using the Pinchpoint Mapper module of the Linkage Mapping plug-in.

Ecological barrier points are areas where species movement between habitat patches is restricted, and their removal can increase connectivity between ecological sources [20], so they should be ecologically restored. Ecological barrier points can be identified by calculating the magnitude of the current recovery value after removing the points. This identification can be accomplished either by selecting the full ecological barrier points that have an impact on the area's ecological flow operation, or by locating the areas that are partially, but not completely, blocked [23,24]. The ecological barrier point locations in the ecological corridor are identified using the Barrier Mapper module in this study. The model is set to "Maximum" calculation mode, the maximum exploration radius and the minimum exploration path are both set to 200 m, and the iteration radius is set to 50 m.

Discovering and protecting ecological corridors can help with ecological restoration, and restoring ecological "pinch points" and ecological barrier points can help with ecosystem function. The two work together to provide incredibly important protection. Since ecological barrier points are substantial areas that block biological flow, their rehabilitation can greatly increase landscape connectedness. Ecological "pinch points" and "barrier points" on the created ecological corridors are selected as the main ecological "pinch points" for ecological preservation, as well as the primary ecological barrier points for ecological restoration, and they are given priority for protection and restoration.

## 3. Results

### 3.1. Spatial Distribution of the Ecological Sources

This study extracts the distributions of forest land, garden land, and grassland from the current land use/cover of the Third National Land Survey, and uses them as ecological land and preliminary patches for the selection of ecological sources. Ecological land is more widespread in the western part of the country than in the eastern urban area. The northern mining region and the southern desert region are the areas with some dispersed ecological land (Figure 3). We then extracted 34,250 patches of ecological land, totaling 2062.65 km$^2$, with the largest patch being 121.19 km$^2$ and making up 5.78% of the total area. The patches of ecological land that have been extracted are generally dispersed, and the top 1000 patches in terms of the area have a total area of 75.05%. Figure 3 shows that some townships in the southwest, including Barren Town, Michae Town, and Shaputul Town, are highly fragmented and sparse. These townships are strongly tied to the intense local human and economic activity. In some towns and farms in the northeast, relatively less

human activity has resulted in large and concentrated patches of ecological lands, such as Ichiban Farm, Yingbari Town, and Yudyklik Town.

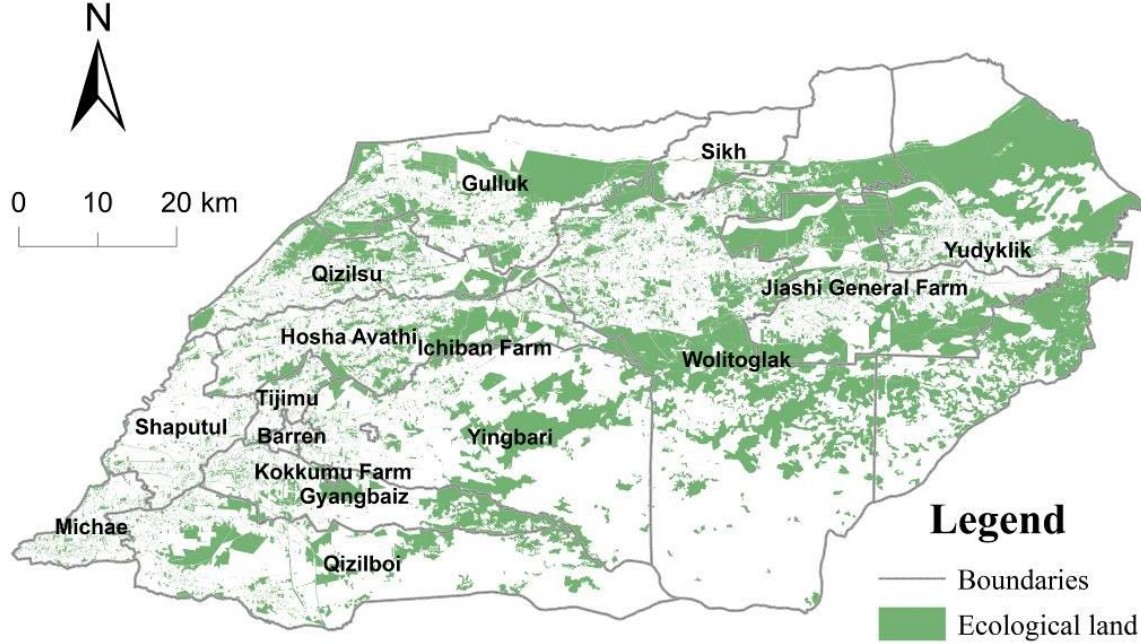

**Figure 3.** Distribution map of ecological land.

The landscape pattern index was calculated using Fragstats4.2 software (University of Massachusetts Amherst, Amherst, MA, USA) at various granularity levels. The nearby ratio and aggregation degree first grew and gradually fell, and then unexpectedly increased when the granular size was 1400 m. The group percentage, patch density, and average proximity also progressively stabilized with increasing granular size. The connection index exhibits an increasing and, subsequently, a declining pattern, peaking at a value of 1400 m for granular size. The cohesion index, the number of fractional dimensions, and the number of maximum component patches all exhibit erratic fluctuations; the cohesion index and the number of fractional dimensions, for example, had a general decreasing tendency, while the number of maximum component patches, on the other hand, had a general upward trend (Figure 4).

The determination index of the landscape pattern was computed using principal component analysis in SPSS25 to identify the best components. Based on earlier research [34], the nine indicators were chosen, and the average proximity distance and the number of sub-dimensions were calculated by taking the opposites of those quantities. The principle of principal component extraction was based on the cumulative contribution rate >80% and eigenvalue >1, and the principal component function was computed in SPSS. The principal component analysis' functional expression was obtained as:

$$Z_1 = 0.3941X_1 + 0.3942X_2 - 0.3897X_3 + 0.4071X_4 - 0.1507X_5 + 0.2914X_6$$
$$-0.2720X_7 + 0.4042X_8 - 0.1398X_9 \tag{5}$$

$$Z_2 = -0.0061X_1 + 0.0043X_2 - 0.0372X_3 - 0.0857X_4 - 0.6982X_5 + 0.4084X_6$$
$$+0.3504X_7 - 0.1254X_8 + 0.4447X_9 \tag{6}$$

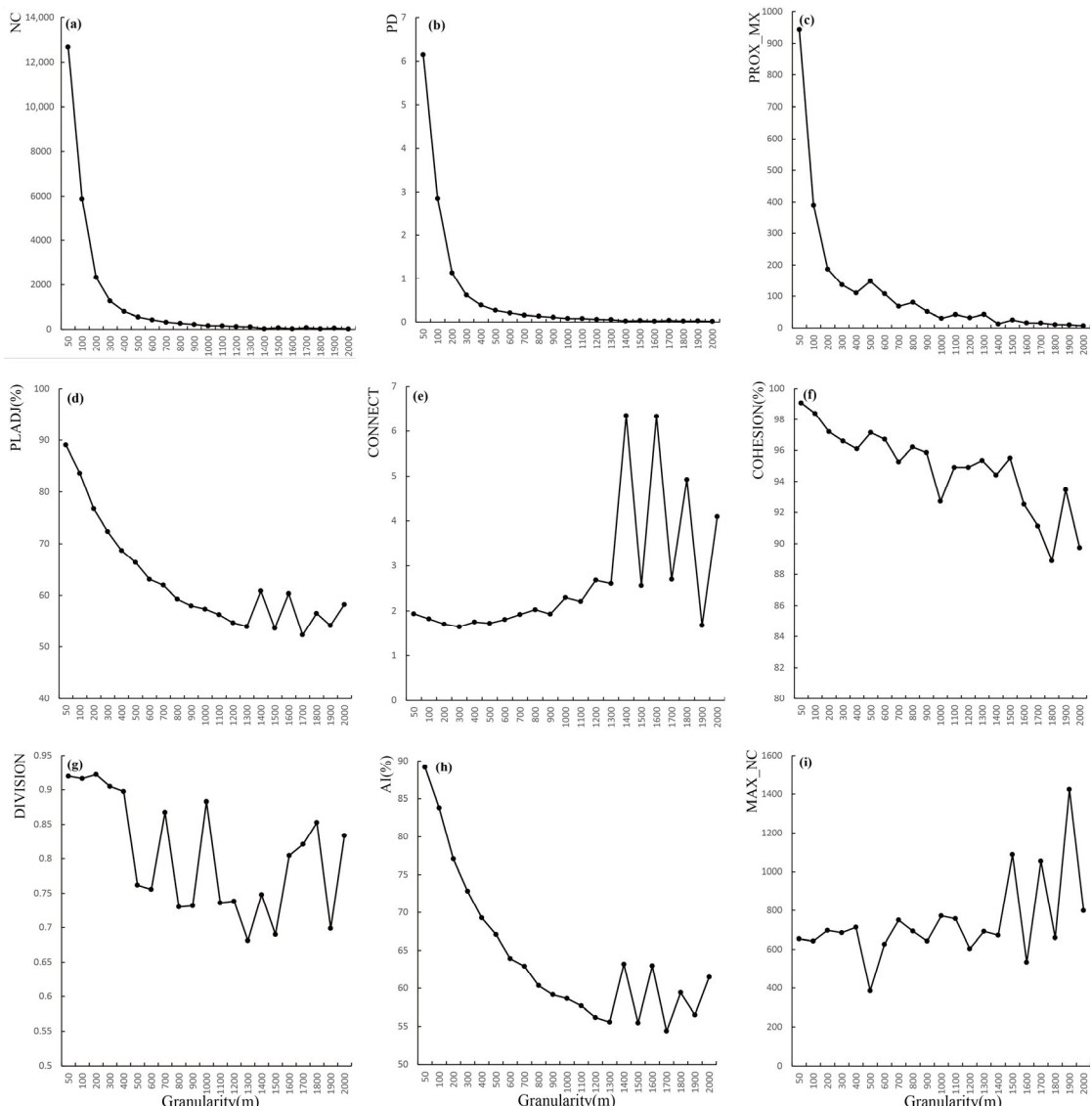

**Figure 4.** Landscape structure index changes with granularity. (**a**) The variation of Number of Patch Components (NC); (**b**) Patch Density (PD); (**c**) Maximum Number of Component Plagues (Max NC); (**d**) Proximity Mean Distance (PROX_MN); (**e**) Proportion of Like Adjacencies (PLADJ); (**f**) Connectivity Index (CONNECT); (**g**) Patch Cohesion Index (COHESION); (**h**) Aggregation Index (AI); (**i**) and Landscape Division Index (DIVISON) with granularity.

The nine indicators mentioned above were subjected to principal component analysis, and two functions—principal component 1 and principal component 2—were produced. Principal component 2 has higher values of average proximity ratio and sub-dimensionality, which, to some extent, reflect the fragmentation of landscape structure, and principal component 1 has higher values of patch density, connectivity, cohesion, and aggregation than principal component 2, better reflecting the connectivity of landscape structure components. The values of the primary components fluctuate together with the granularity change, and the landscape structure is thought to be in a more cohesive state overall when the changing values buck the trend of change [34]. It is assumed that overall connectedness is higher when the value of principal component 1 changes abruptly, and that overall fragmentation is larger when the value of main component 2 changes abruptly. According to Figure 5, the change in the component structure of the landscape is qualitatively different at the size of 1400 m. This point is the optimal structure for the ecological source, because its value

is higher than that of other nearby dimensions. Thus, the size of 1400 m of the landscape structure serves as a suitable benchmark for the selection of the ecological source.

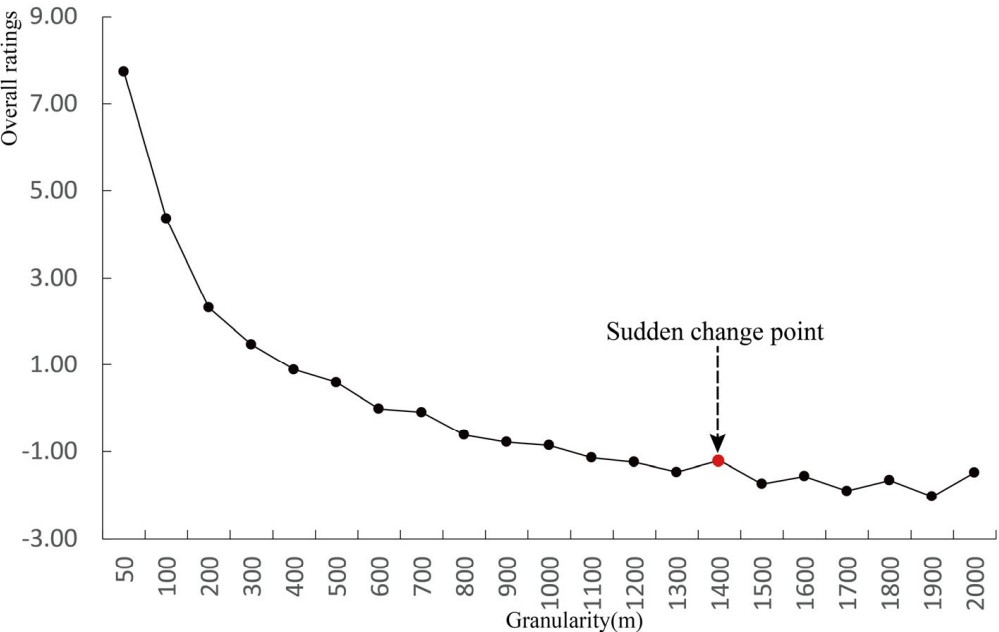

**Figure 5.** The comprehensive score of landscape structure component stability. The ideal size appears at the sudden change point.

The InVEST habitat quality model was used to determine the integrated habitat quality of Jiashi county, in order to generate a distribution map of habitat quality and to characterize the spatial pattern of biodiversity in Jiashi county. As shown in Figure 5, the level of habitat diversity is represented by the shade of green; the darker the color, the better the biodiversity. The analysis shows that Jiashi's ecological land has a decent overall quality, with a mean value of 0.54 and a range of 0.7 to 0.4. The overall pattern is high in the central and eastern parts of the county and low in the western, southern, and northern parts of the county, with the central part of Jiashi county being an area of high habitat quality.

The principle of source selection from ecological land is to comprehensively consider patch connectivity and habitat quality. Based on the granularity inverse method and the habitat quality model, ecological land with higher connectivity and habitat quality was selected as the ecological source. The first step was to choose the ideal granular size using the granularity inverse method. Higher connectivity can be achieved by choosing a source with a granular size of 1400 m, which is the ideal size. Ecological sources need to take into account high habitat quality while taking into account connectivity, and use regional statistics to calculate granular patches with high habitat quality (Figure 6a). These granular patches were used to inversely select patches of ecological land, and, finally, 36 ecological sources were selected. The selected ecological source area is 1331.13 km$^2$ in total, of which grassland is the main ecological source, accounting for 65.46% of the ecological source area. The ecological sources exhibit a spatial pattern of higher levels in the east and lower levels in the west, north, and south, as shown in Figure 6b.

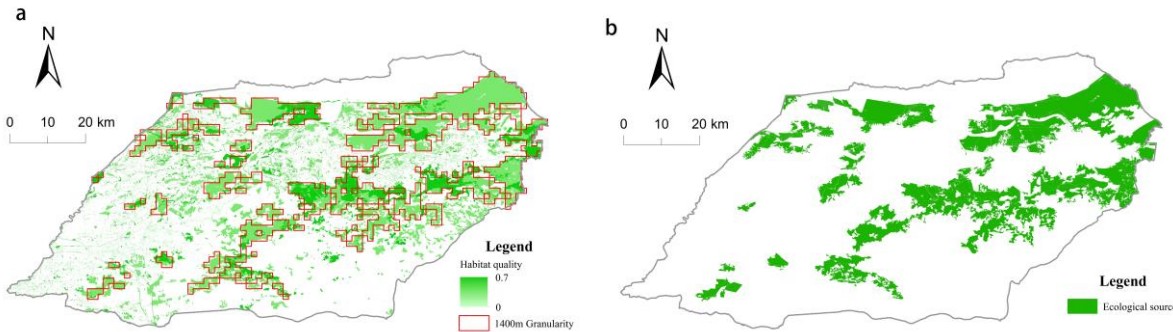

**Figure 6.** Spatial distributions of the selected ecological sources in Jiashi county. (**a**) The habitat quality in the ecological land, and the range framed by the red line are granular patches with high habitat qual-ity; (**b**) Shows the final selected ecological source.

### 3.2. Identification of Resistance Surface and Ecological Corridor

Land use data, nighttime light data, terrain, and geomorphology data are integrated to generate the comprehensive resistance surface of Jiashi county. As shown in Figure 7, the high-value resistance areas in Jiashi county are concentrated in the center of Barren town, as well as other townships and the high mountain area in the north of Jiashi county, which is mainly damped by urban construction land, traffic roads, and high mountains. Ecological corridors are identified in ArcCatalog 10.2, utilizing the Linkage Mapper plug-in to connect ecological sources based on the integrated resistance surface. According to Figure 7, 52 ecological corridors totaling 316.30 km in length were selected, with the longest ecological corridor measuring 34.63 km. Jiashi county's ecological corridors were mostly concentrated in the west, with the longest ecological corridor located there. The number of corridors in the northeast was also the highest, which was closely tied to the region's more sparsely distributed ecological source in the northwest. Compared with the west, the corridors in the east are shorter and fewer, because the patches in the eastern ecological sources land are larger and fewer.

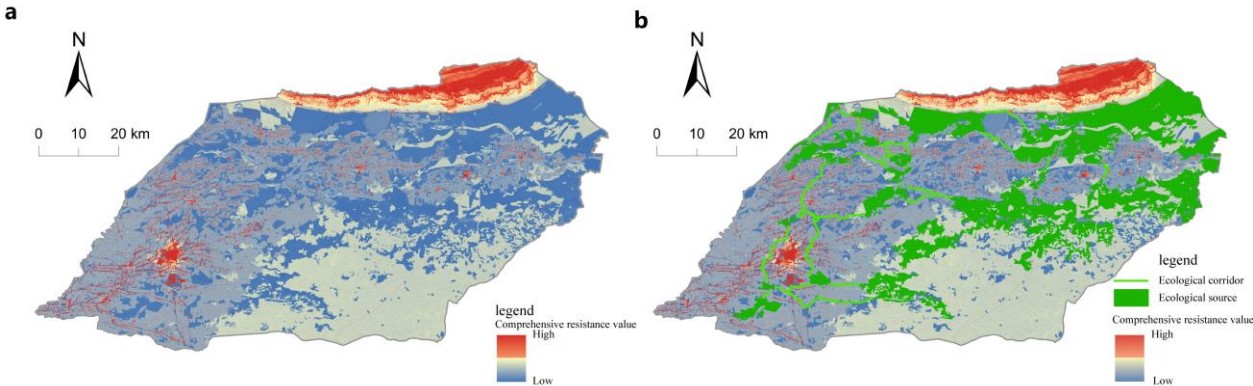

**Figure 7.** Resistance surface and ecological corridor diagram. (**a**) Shows the resistance surface, where the resistance value decreases gradually from red to blue. (**b**) Shows the distribution of ecological sources and ecological corridors on the resistance surface.

### 3.3. Identification of Key Areas for Ecological Restoration of the Territory Land Space

Using the circuit theory, the ecological source of Jiashi county was injected with current to obtain the current distribution map (Figure 8). The areas with strong point currents were retrieved using the natural break approach, and the selection of "pinch points" larger than 10 km² were selected to ensure the connectivity of the regional ecosystem [35]. The identified ecological corridors were chosen as source layers in ArcGIS10.2 based on their location, and 164 of the 215 "pinch points", with a combined size of 16.53 km², were retrieved. From 215 "pinch points", the crucial "pinch points" of the corridors were

identified. There were found to be 164 significant ecological "pinch places", covering a total of 15.13 km$^2$.

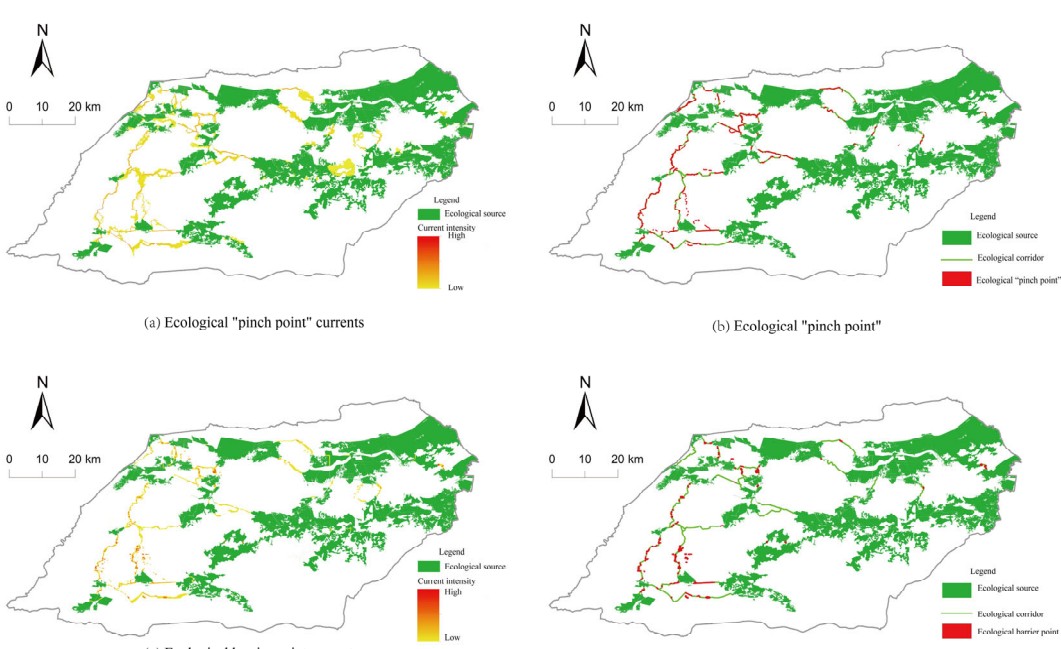

**Figure 8.** Identification of ecological key points.

As shown in Figure 8b, the red area is an ecological "pinch point", which is mainly distributed in the east of Jiashi county, indicating that the east of Jiashi county must be safeguarded more actively and given more attention during the planning process. The Linkage Mapper Toolkit plug-Barrier in's Mapper was used to extract the ecological barrier points based on the circuit theory. A total of 69 major ecological barrier points were selected near the ecological corridor, blocking biological movement, with a total area of 14.57 km$^2$ out of the total 85 ecological barrier points that were discovered, covering a total area of 16.76 km$^2$(Figure 8d).

*3.4. Optimization of the Spatial Ecological Pattern of Territory Land Space Based on "Source-Corridor"*

The break points were determined as focal points for ecological restoration after the main road data and the selected corridor data were intersected to produce them. Table 4 demonstrates how the chosen ecological source and the identified key "pinch points" are set as ecological restoration protection areas, as well as the areas with high patch connectivity but poor regional habitat quality. Identified key barrier points are set as ecological restoration improvement areas to create the ecological restoration pattern in land space. In the eastern part of Jiashi county, at the intersections of roads and corridors, 14 ecological break points are found as illustrated in Figure 9. In order to promote the passage of species between different sources, ecological channels should be constructed at ecological break points. The eastern area of Jiashi county contains the majority of the county's ecological corridors. The ecological restoration and improvement area is spread throughout the entire territory of Jiashi county, which is both comparatively small and excellent. It should, primarily, be enhanced and improved, while areas with high patch connectivity but poor habitat quality should be enhanced and improved with adequate funding. The restoration and enhancement of ecological barriers in the ecological restoration and improvement areas can considerably improve ecosystem functions. Ecological restoration reserves are dispersed throughout Jiashi county and take up a sizable amount of space, making them a crucial protected area. While protecting the ecological source, strictly enforcing the land use control system, and strengthening oversight of remote sensing land enforcement, it

is necessary to protect the ecological "pinch points" in the ecological restoration reserves, and to engage in restoration. Ecological restoration in Jiashi county is divided into three modes, including natural restoration, manual intervention, and control protection. Diverse modes target various land types. Natural restoration focuses on the areas with low habitat quality in the existing ecological land; manual intervention focuses on the restoration of identified key areas that hinder biological movement; and control and protection focuses on ecological sources and ecological "pinch points".

**Table 4.** Classification of ecological land space restoration types.

| Type | Content | Restoration Strategy |
|---|---|---|
| Key points of ecological restoration | Ecological break points | Building biological pathways |
| Ecological restoration reserve | Ecological "pinch points", identified ecological source | Upgrading and improvement; strict implementation of land use control system as well as strengthening supervision of remote sensing land enforcement; natural restoration |
| Ecological restoration and improvement area | Ecological land of low habitat quality, ecological barrier points | Prioritizing protection and investing in restoration; manual control vs. controlled protection |

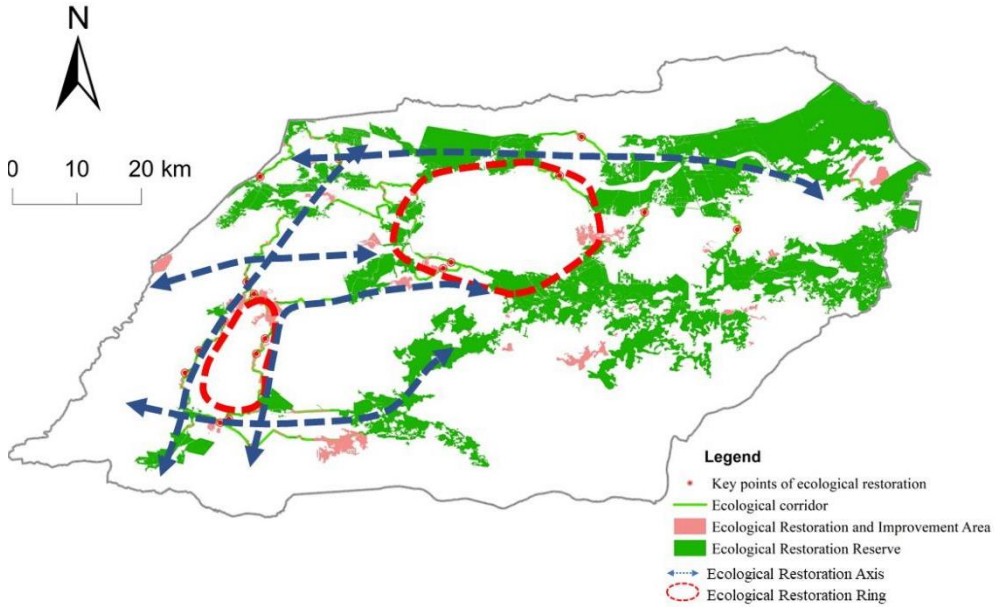

**Figure 9.** Spatial optimization pattern for the ecological restoration in Jiashi county.

We presented the "5 + 2 + N" design for spatial optimization for the ecological restoration of "five axes, two rings, and multiple regions" in Jiashi county based on the identified ecological corridors and key ecological restoration areas. "Five-axis, two-ring, multi-zone" (5 + 2 + N) is a guide for restoration and an important part of the ecological restoration of the country. The five axes are defined according to the combination of main ecological corridors, river systems, and land use/cover types, including the axis belt constructed along the key ecological corridors. They are dispersed over Jiashi county, showing a pattern of "two horizontal and three vertical", which is used to guide the construction of ecological corridors. The two rings are mainly distributed in the outer layer of the areas with intensive human activities in the east and middle of Jiashi county, including the ecological corridor around the central urban area and the ecological corridor around Kashgar River Xikel Reservoir. The construction of circular corridors in these areas can enhance ecosystem performance while also benefiting society and the local populace spiritually. The multi-regions

include multiple identified ecological restorations and protected areas, as well as ecological restoration and improvement areas.

## 4. Discussion

This study adopted the ecological security pattern paired with circuit theory to identify ecological corridors and key areas for ecological restoration in Jiashi county based on the "source-corridor" paradigm. The process of identifying the ecological source took into account both the integration of the connectedness of ecological sources and other landscape characteristics, which improves the plausibility of the ecological source's landscape structures [36]. Additionally, the coupling of the ecological source identification granularity inverse method lowered the fragmentation of parcels. The InVEST habitat quality model was finally able to identify 36 ecological sources in Jashi county after determining that the ideal granularity for identification is 1400 m. Each patch within the same granularity range was treated as a separate source. It can be found that this strategy is more logical than the previous research, which chose the final ecological source based on the sources' areas and connectivity percentages [37]. Compared to the first approach, this version is more practical and reasonable. Regarding the resistance surface, we used land use data, nighttime lighting data, and topography to construct a resistance surface. The resistance value was high, primarily in urban and mountainous areas, and the resistance surface was primarily made up of construction land for cities, roads for traffic, and mountains. This research also describes how the current resistance surface is distributed in Jiashi county. The identified ecological corridors connect numerous ecological sources and are dispersed throughout Jiashi county, which is crucial for the migration of species. The restoration of these places is extremely important to local ecological protection. The key areas of ecological restoration are mainly ecological "pinch points", ecological barrier points, and ecological break points. The restoration of these areas is of great significance to the regional ecosystem. Therefore, strengthening the restoration of key ecological areas and the construction of ecological corridors will have an important reference value for ecological restoration planning. However, due to different methods of source identification and resistance surface construction, the results of the identified "source corridor" will also be different [9]. In terms of source area identification, this paper comprehensively chose the ecological sources from the ecological land in the land use data, which were taken from the three national land surveys based on the biodiversity and landscape characteristics. This study was conducted without taking into account the vegetation coverage, water, soil conservation, the supply and demand relationship of ecosystem services, or the sensitivity of ecological source areas of various land uses in Jiashi county. Using the information on human activities and geography, we corrected the resistance surface according to land use/cover type. Factors such as the density of the roads and the amount of vegetation cover were not taken into account. The ecological corridors and key areas of ecological restoration, identified based on different priorities and purposes, are different. Due to their natural and social conditions, each region has distinctive requirements for the creation of ecological corridors and restoration areas. This study, which is based on the Third National Land Survey, aims to implement the follow-up work on regional protection and restoration in a specific way. In order to advance the ecological restoration work in China, this study would offer a new approach and route for the identification of ecological corridors and key areas of ecological restoration, which can serve as an ecological restoration strategy.

The habitat quality module of InVEST can, to some extent, reflect biodiversity in terms of the choice of research methodologies and data. However, it mostly depends on Ecological Security Pattern data, which cannot adequately represent biodiversity. Field survey data can accurately represent biodiversity, but Jiashi county lacks species distribution data, and it is challenging to collect a significant amount of species data. Therefore, this study substituted habitat quality for biodiversity. In addition, rivers and lakes were not chosen as sources; due to the study area's location along the southern border, the area of rivers is very small, and there are mainly seasonal water breaks. However, water connectivity is an

important research topic in ecological restoration, and the connectivity of water bodies can be examined in greater detail in the future. This study was based on the current state of the entirety of Jiashi county in order to identify the ecological corridor. Due to the lack of data, there is no in-depth study based on land use dynamics and multi-species corridors. The identification of ecological corridors based on multi-species and different scenarios of future land use will be an important research direction in the future.

Additionally, the study of ecological corridors based on expected climate change has also begun to draw attention [38–40]. The verification of ecological corridors is equally important. Recent studies have used GPS technology to verify ecological corridors on a global scale [41]. Validation of ecological corridors at the regional, national, and city/county scales is also extremely necessary. This verification work will promote more in-depth research on ecological corridors.

## 5. Conclusions

Based on the ecological security pattern paradigm of the "source-corridor", this study combined the InVEST Habitat Quality model with the granularity inverse method to identify the ecological sources. This method combines biodiversity and general connectivity. Ecological "pinch points", ecological barrier points, ecological break points, etc., were then diagnosed using the minimal cumulative resistance model and circuit theory, and, finally, the main ecological restoration areas in the study area were determined. The study found that: (1) Based on ecological sources and resistance surfaces, 52 ecological corridors totaling 316.30 km have been identified, covering a source area of 1331.13 km$^2$, and 15.13 km$^2$ of significant ecological "pinch points" have been discovered, mostly in the eastern section of Jiashi county. With a total area of 14.57 km$^2$, the eastern portion of Jiashi county is home to the majority of the county's 69 critical ecological barrier locations. The analysis of the habitat quality of Jiashi county's ecological source land reveals that the ecological land's overall quality is generally good, with an average value of 0.54 and a strong natural resource foundation. Ecological restoration in Jiashi county should be based on natural restoration, supplemented by artificial restoration, and focused on protecting the eastern part of Jiashi county in the planning process. (2) There are 164 ecological "pinch points" that need to be rehabilitated; their predominant land uses are other grasslands, wetlands, shrublands, etc. In Jiashi county, 69 ecological barriers need to be removed. The primary forms of land usage are watering land and homesites. The local ecological restoration process focuses on encouraging rural landscape planting and the development of characteristic agriculture, forestry economies, etc. As a result, such ecological restoration can prioritize the growth of fruit trees beneath home sites, and forests and grains beneath agricultural land. Our findings can serve as a sound scientific foundation for the creation of plans for ecological restoration in the territory land space.

**Author Contributions:** Conceptualization, J.D., Y.'e.C. and X.F.; methodology, J.D.; software, J.D., C.L. and Y.L. (Yinyin Liang); validation, J.D., Y.'e.C. and X.F.; formal analysis, J.D., Y.'e.C. and X.F.; investigation, J.D., Y.'e.C. and C.L.; resources, J.D., Y.L. (Yinyin Liang) and Y.'e.C.; data curation, J.D. and X.F.; writing—original draft preparation, J.D. and Y.'e.C.; writing—review and editing, C.L., Y.'e.C., X.F., Y.L. (Yijing Liu) and C.Z.; visualization, J.D.; supervision, X.F. and Y.'e.C.; project administration, Y.'e.C.; funding acquisition, Y.'e.C. and C.L. All authors have read and agreed to the published version of the manuscript.

**Funding:** This research study was supported by National Natural Science Foundation of China (NSFC) (NO. 41971060) and Shanghai Normal University, under the "2022 Construction of High-level Local Universities First-class Graduate Education Project".

**Institutional Review Board Statement:** Not applicable.

**Informed Consent Statement:** Informed consent was obtained from all subjects involved in the study.

**Data Availability Statement:** The data presented are available upon request from the corresponding author.

**Conflicts of Interest:** The authors declare no conflict of interest.

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
