# Peer review of "Identification of Key Areas for Ecosystem Restoration Based on Ecological Security Pattern"

_sustainability, doi:10.3390/su142315499_

Round 1

Reviewer 1 Report

General comments:

Comments on the manuscripts (MS) “Identification and Diagnosis of Territorial Key Areas for Ecological Restoration in China based on ‘Source-Corridor’ Paradigm--A case study of Jiashi County” by Jiaquan Duan et al.

Ecological security pattern is an important spatial solution for territorial ecological restoration. Regional identification of key areas for ecological restoration in territorial ecosystem is an important for spatial planning on ecological restoration. This manuscript presents this important topic regarding how to identify priority areas for land restoration based on based on ecological security pattern. It addresses important topics of interest to the readers of the journal with sound data and methodologies. It can provide useful information for ecological restoration planning in this region. However, I have the following specific suggestions for improvement.

1)    The title of this MS is not attractive and too long. I recommend that authors could change it into “Identification of priority areas for land restoration based on ecological security pattern”

2)    The English of this article still needs to be greatly improved. Some nouns are translated into two different words. Such as In line 12 “granularity inversion mothod” and in line 22 “particle size inversion method”.  Please ask a native English speaker to help polish the paper.

3)    In line 112, “The research method includes three parts, i.e.”. what means “i.e.”. what do you mean in there?

4)     Why is biodiversity assessed using the InVEST model? More explanation is suggested in this regard.

5)    Please provide specific explanations on how the threat data and sensitivity data come from inTables 2 and 3

6)    The author is suggested to add more content in the discussion section

7)    In line 162, the statement "calculating the granularity of 21 scales such as 50m, 100m, 200m, 300m, 400m, 500m, 600m, 700m, 800m, 900m, and 100m" is wrong. Where did you select the 21 scales?

Author Response

Comments on the manuscripts (MS) “Identification and Diagnosis of Territorial Key Areas for Ecological Restoration in China based on ‘Source-Corridor’ Paradigm--A case study of Jiashi County” by Jiaquan Duan et al.

Ecological security pattern is an important spatial solution for territorial ecological restoration. Regional identification of key areas for ecological restoration in territorial ecosystem is an important for spatial planning on ecological restoration. This manuscript presents this important topic regarding how to identify priority areas for land restoration based on based on ecological security pattern. It addresses important topics of interest to the readers of the journal with sound data and methodologies. It can provide useful information for ecological restoration planning in this region. However, I have the following specific suggestions for improvement.

1)    The title of this MS is not attractive and too long. I recommend that authors could change it into “Identification of priority areas for land restoration based on ecological security pattern”

Response:Thanks to the editor's comments, after careful consideration, we decided to change the article to " Identification of Priority Areas for Ecosytem Restoration Based on Ecological Security Pattern"

2)    The English of this article still needs to be greatly improved. Some nouns are translated into two different words. Such as in line 12 “granularity inversion method” and in line 22 “particle size inversion method”.  Please ask a native English speaker to help polish the paper.

Response: Thanks to the reviewers for their suggestions, we read the full text and consulted experts to revise the entire article. Inappropriate sentences have been removed, and new sentences have been added to make the article easier to understand. (E.g., line8-22 23 151-156 350-352 553 749 768-789 791-793 874-876) We are very sorry for such a problem In line 12 “granularity inversion method” and in line 22 “particle size inversion method”, it is an omission of our work. We've modified it so that it works.

3)    In line 112, “The research method includes three parts, i.e.”. what means “i.e.”. what do you mean in there?

Response: We looked up the information on Google. i.e., stands for id est, which means "that is"

4)     Why is biodiversity assessed using the InVEST model? More explanation is suggested in this regard.

Response: Thanks to the reviewers for their comments. Referring to relevant literature [1,2], InVEST habitat quality can reflect biodiversity to a certain extent. At the same time, considering the lack of species survey data in Xinjiang, it is difficult to obtain data on related species. For the time being, only the calculation results of the habitat quality model in InVEST can be used to replace the analysis results. At the same time, we also discussed this issue in the Discussion section in line 1093-1098.

Reference:

  1. Nelson, E.; Mendoza, G.; Regetz, J.; Polasky, S.; Tallis, H.; Cameron, D.; Chan, K. M.; Daily, G. C.; Goldstein, J.; Kareiva, P. M., Modeling multiple ecosystem services, biodiversity conservation, commodity production, and tradeoffs at landscape scales. Frontiers in Ecology and the Environment 2009,7, (1), 4-11.

  1. Zhang, L.; Peng, J.; Liu, Y.; Wu, J., Coupling ecosystem services supply and human ecological demand to identify landscape ecological security pattern: A case study in Beijing–Tianjin–Hebei region, China. Urban Ecosystems 2017,20, (3), 701-714.
  2. Liu, H. J.; Niu, T.; Yu, Q.; Yang, L. Z.; Ma, J.; Qiu, S.; Wang, R. R.; Liu, W.; Li, J. Z., Spatial and temporal variations in the relationship between the topological structure of eco-spatial network and biodiversity maintenance function in China. Ecological Indicators 2022,139.

5)    Please provide specific explanations on how the threat data and sensitivity data come from in tables 2 and 3

Response: Many thanks to the reviewers for their suggestions. The data on threat sources and susceptibility are mainly obtained by comprehensive consideration concerning relevant literature and InVEST guidelines [1, 2].

Reference:

  1. Liu Chunfang; Wang Chuan, Temporal and spatial evolution characteristics of habitat quality in the loess hilly area based on land use change: a case study of Yuzhong County. Acta Sinica Ecology2018,38, (20), 7300-7311.
  2. Zhang Xueru; Zhou Jie; Li Mengmei, Temporal and Spatial Variation Analysis of Regional Habitat Quality Based on Land Use Pattern Reconstruction. Acta Geographica 2020,75, (01), 160-178.

6)    The author is suggested to add more content in the discussion section

Response: Thanks to the reviewers for their comments, We are also aware of our lack of outlook on recent research in the Discussion section. The results of the latest research published in Science, we have added some new prospects for future research in the discussion section. See line 1107-1114 for details

7)    In line 162, the statement "calculating the granularity of 21 scales such as 50m, 100m, 200m, 300m, 400m, 500m, 600m, 700m, 800m, 900m, and 100m" is wrong. Where did you select the 21 scales?

Response: Thank you very much for the questions pointed out by the reviewers. In response to the comments of the reviewers, we have revised the relevant statements in the part where the statement is wrong. (line:954) The landscape pattern index changes continuously with the granularity. According to the granularity inversion method, to find out the granularity of its mutation, the more granularity, the better, but the more granularity, the less obvious the mutation of granularity. This study is based on the county scale, and considering the small size of the study area, the granularity starts from 50m. We also calculated the values of the average proximity distance and connectivity index after 2000m granularity, and the average proximity distance and connectivity index were both 0 after the particle size was greater than 2000m. Referring to the research of Lu Yu et al., about 21 scales can better reflect the trend of granularity mutation. Therefore, this study calculated the particle size of 21 scales such as 50m, 100m, 200m, 300m, 400m, 500m, 600m, 700m, 800m, 900m, and 100m.

Reviewer 2 Report

It is judged to be an good paper studied so that the results can be used in practice by presenting methods and procedures for ecological restoration for Jiashi County. However, the resolution of the picture (Figs. 4, 6, 7, 8) is low, so it is impossible to understand, and the paper is organized based on the research report, so it is somewhat awkward. Also, there are many mis-subscribers or mixed terms found across the board.

 I do not understand what theoretical background the basis of 'five-axis, two-ring, multi-are' presented in line 428. In addition, the meaning of the 'Source-Corridor' presented in the title of the paper should be clearly presented in this study.

 It is determined that 'patch' in Table 1, 'particle size' in Figure 4, and 'Graularity' in Figure 4 have the same meaning. If it's the same meaning, you have to match the terms.

 I could not understand the meaning of 'Gendenland' and 'Bulit-upland' presented in the legend in Figure 1. Specifically, what kind of condition does it mean?

 The title in Figure 2 seems to be more appropriate for the 'flow chart' than the 'technical road map' presented.

 In Table 1, 'PROX-MN' differs from the variables presented in the expression. In addition, the variables presented in 'COHESION' are also required to be corrected to typos.

 'garden land' shown in line 288 is not shown in Figure 1. Please check if 'Gendenland' in Figure 1 is 'garden land'.

Author Response

It is judged to be an good paper studied so that the results can be used in practice by presenting methods and procedures for ecological restoration for Jiashi County. However, the resolution of the picture (Figs. 4, 6, 7, 8) is low, so it is impossible to understand, and the paper is organized based on the research report, so it is somewhat awkward. Also, there are many mis-subscribers or mixed terms found across the board.

Response: Thanks for the editor's suggestion. For the incomprehensible problem that the resolution is too low, we modify the resolution of the image. In the meantime, thanks for your suggestion about mis-subscribers or mixed terms found across the board. In response to this problem, we have carefully checked the full text and revised mis-subscribers or mixed terms found across the board. Many of our revisions can be seen in the text, mainly modifying the terminology for consistency. At the same time, to ensure that the terms are correct, we also refer to the relevant literature for verification.

Reference:

  1. Zhang, L.; Peng, J.; Liu, Y.; Wu, J., Coupling ecosystem services supply and human ecological demand to identify landscape ecological security pattern: A case study in Beijing–Tianjin–Hebei region, China. Urban Ecosystems 2017,20, (3), 701-714.
  2. Peng, J.; Zhao, S.; Dong, J.; Liu, Y.; Meersmans, J.; Li, H.; Wu, J., Applying ant colony algorithm to identify ecological security patterns in megacities. Environmental Modelling & Software 2019,117, 214-222.

 I do not understand what theoretical background the basis of 'five-axis, two-ring, multi-are' presented in line 428. In addition, the meaning of the 'Source-Corridor' presented in the title of the paper should be clearly presented in this study.

Response: Thanks for the editor's suggestion. We are very sorry for not being able to make ‘five-axis, two-ring, multi-are' clear, and we find that it can’t match its explanation. So, we change the content in line 954 to match its explanation in lines955-965. ‘five-axis, two-ring, multi-zone is also a guide for restoration and an important part of the ecological restoration of the country. For a better understanding of the reader, we also explain it in more depth in line953-955. In response to the unclear meaning of the "source-corridor" paradigm, we have added a description of the "source-corridor" paradigm, see line 116-117 for details.

It is determined that 'patch' in Table 1, 'particle size' in Figure 4, and 'Graularity' in Figure 4 have the same meaning. If it's the same meaning, you have to match the terms.

Response: Thanks for the editor's suggestion. we determined that 'patch' in Table 1. And 'particle size' in Figure 4 and 'Graularity' in Figure 4 have the same meaning. We are very sorry for such a problem, it is an omission of our work. We've modified it so that it works.

 I could not understand the meaning of 'Gendenland' and 'Bulit-upland' presented in the legend in Figure 1. Specifically, what kind of condition does it mean?

Response: Thanks to the editor for pointing out the problem. ‘Gendenland’ in the text is a typo in our writing process and 'Bulit-upland' is a mistake.  We checked relevant literature and changed 'Gendenland' to "garden land" in Figure 1, and changed 'Bulit-upland' to "Construction land". Thanks for the editor's suggestion.

 The title in Figure 2 seems to be more appropriate for the 'flow chart' than the 'technical road map' presented.

Response: Thanks for the editor's suggestion. We modified line 175 in the text.

 In Table 1, 'PROX-MN' differs from the variables presented in the expression. In addition, the variables presented in 'COHESION' are also required to be corrected for typos.

Response: Thanks for the editor's suggestion. we have modified the expression of the formula, see line 478 for details

 'garden land' shown in line 288 is not shown in Figure 1. Please check if 'Gendenland' in Figure 1 is 'garden land'.

Response:Thanks for the editor's suggestion. To make the text in the following text correspond, we have modified the relevant content in Fig.1.